# Spatially Resolved Molecular Characterization of Noninvasive Follicular Thyroid Neoplasms with Papillary-like Nuclear Features (NIFTPs) Identifies a Distinct Proteomic Signature Associated with RAS-Mutant Lesions

**DOI:** 10.3390/ijms252313115

**Published:** 2024-12-06

**Authors:** Vanna Denti, Angela Greco, Antonio Maria Alviano, Giulia Capitoli, Nicole Monza, Andrew Smith, Daniela Pilla, Alice Maggioni, Mariia Ivanova, Konstantinos Venetis, Fausto Maffini, Mattia Garancini, Angela Ida Pincelli, Stefania Galimberti, Fulvio Magni, Nicola Fusco, Vincenzo L’Imperio, Fabio Pagni

**Affiliations:** 1Proteomics and Metabolomics Unit, Department of Medicine and Surgery, University of Milano-Bicocca, 20900 Monza, Italy; vanna.denti@unimib.it (V.D.);; 2Department of Medicine and Surgery, Pathology, Center of Digital Medicine, University of Milano-Bicocca, Fondazione IRCCS San Gerardo dei Tintori, Via Cadore 48, 20900 Monza, Italy; 3Bicocca Bioinformatics Biostatistics and Bioimaging Research Centre—B4, Department of Medicine and Surgery, University of Milano-Bicocca, 20900 Monza, Italystefania.galimberti@unimib.it (S.G.); 4Biostatistics and Clinical Epidemiology, Fondazione IRCCS San Gerardo dei Tintori, 20900 Monza, Italy; 5Department of Pathology and Laboratory Medicine, European Institute of Oncology IRCCS, 20141 Milan, Italy; 6Fondazione IRCCS San Gerardo dei Tintori, 20900 Monza, Italy; mattia.garancini@irccs-sangerardo.it (M.G.);; 7Department of Oncology & Hemato-Oncology, University of Milan, 20122 Milan, Italy

**Keywords:** thyroid neoplasms, NIFTP, MALDI-MSI, proteomics, NGS, *RAS* mutations

## Abstract

Follicular-patterned thyroid neoplasms comprise a diverse group of lesions that pose significant challenges in terms of differential diagnosis based solely on morphologic and genetic features. Thus, the identification of easily testable biomarkers complementing microscopic and genetic analyses is a highly anticipated advancement that could improve diagnostic accuracy, particularly for noninvasive follicular thyroid neoplasms with papillary-like nuclear features (NIFTPs). These tumors exhibit considerable morphological and molecular heterogeneity, which may complicate their distinction from structurally similar neoplasms, especially when genetic analyses reveal shared genomic alterations (e.g., *RAS* mutations). Here, we integrated next-generation sequencing (NGS) with matrix-assisted laser desorption/ionization mass spectrometry imaging (MALDI-MSI) to perform a proteogenomic analysis on 85 NIFTPs (n = 30 *RAS*-mutant [*RAS*-mut] and n = 55 *RAS*-wild type [*RAS*-wt]), with the aim to detect putative biomarkers of RAS-mut lesions. Through this combined approach, we identified four proteins that were significantly underexpressed in *RAS*-mut as compared to RAS-wt NIFTPs. These proteins could serve as readily accessible markers in morphologically borderline cases showing *RAS* mutations. Additionally, our findings may provide insights into the distinct pathogenic pathways through which *RAS*-mut and *RAS*-wt NIFTPs arise, highlighting the pivotal role of constitutive RAS–mitogen-activated protein kinase (MAPK) pathway activation in the development and progression of *RAS*-mut tumors.

## 1. Introduction

A well-known heterogeneity can complicate the differential diagnosis of follicular-patterned thyroid lesions, which comprise entities with different clinical course and prognosis [1]. Among them, the noninvasive encapsulated follicular variant of papillary thyroid carcinoma (PTC) tends to show an indolent behavior, with an extremely low risk of recurrence and distant spread. This has led to the proposal of the “non-invasive follicular thyroid neoplasm with papillary-like nuclear features” (NIFTP) nomenclature, with the aim to reduce overdiagnosis and overtreatment [2]. However, several unresolved issues remain. First, inter-pathologist variability in the application of the established diagnostic criteria could hamper the reporting reproducibility, complicating efforts to distinguish such lesions from histologically similar entities. Moreover, the high degree of intra-class heterogeneity, exemplified by the identification of a subgroup of “atypical NIFTPs”, could render the category a “waste bin” for lesions with indeterminate or equivocal features [3]. Interestingly, this heterogeneity is evident not only on a morphological but also on a molecular level, as suggested by the wide variety of mutations in several different genes identified in NIFTP specimens. The most frequently reported alterations involve *RAS*-family genes, while *BRAF* mutations are exceedingly rare, similar to what has been observed in follicular adenomas (FAs) and carcinomas [4,5,6]. This may undermine the usefulness of molecular genetic analyses and add another layer of complexity to the differential diagnostic task, particularly when evaluating cytological specimens. Therefore, there is a pressing need for reliable biomarkers to facilitate the identification of such lesions and to complement the information provided by genetic analyses, especially for cases falling in the “gray zone” of *RAS*-mutated (*RAS*-mut) tumors. To this end, a promising tool in the “-omics” characterization of thyroid neoplasms is matrix-assisted laser desorption/ionization-mass spectrometry imaging (MALDI-MSI) [7]. Previous experiences demonstrated that MALDI-MSI has the potential to dissect the proteomic profile of NIFTPs, enhanced by the simultaneous integration with next-generation sequencing (NGS) data [8]. This in-depth characterization may identify biomarkers that not only support the diagnostic process but also provide valuable insights into the differential pathogenetic pathways through which the various NIFTP subtypes arise. Our group has already demonstrated that *RAS*-wild type (*RAS*-wt) and *RAS*-mut NIFTPs show distinct proteomic signatures, with the former resembling hyperplastic nodules and the latter being enriched for *RAS*-interacting proteins [9]. Thus, we proposed that the two NIFTP subgroups may be intrinsically different in terms of their biology and pathogenesis. We also observed that *RAS*-mut NIFTPs seem to rely on the expression of *RAS*-related proteins to support their growth and progression, suggesting that the identification of a *RAS* mutation in a NIFTP may portend increased tumor cell survival and proliferation and lead to an earlier emergence and a less indolent clinical course. In this hypothetical scenario, the identification of putative biomarkers of *RAS*-mut NIFTPs could be even more valuable since it might also provide useful information to guide patient management.

In the present work, we aimed to further characterize the molecular profile of *RAS*-mut vs. *RAS*-wt NIFTPs by integrating MALDI-MSI with NGS data from a broader and more heterogeneous cohort of patients undergoing thyroidectomy. Our goal was to identify potential proteogenomic targets that could aid in the differential diagnostic process, particularly to rule out a NIFTP diagnosis in morphologically indeterminate lesions with *RAS* mutations. Additionally, we aimed to gain further insight into the specific biological pathways involved in the development of *RAS*-mut and *RAS*-wt NIFTPs to investigate whether intrinsic pathogenetic differences exist between these two NIFTP subcategories, as suggested in our previous study.

## 2. Results

### 2.1. Clinicopathologic Characteristics of the Study Cohort

The clinicopathologic features of the 69 patients enrolled in the study are summarized in Table 1. Detailed clinicopathologic data regarding each patient, as well as data on the molecular alterations detected via next-generation sequencing (NGS) analyses, are reported in Appendix A.

Subjects were stratified according to the *RAS* mutational status of their NIFTPs: in the *RAS*-wt cohort (n = 46 patients, 66.7%), the mean age of the patients was 57.3 ± 11.4 years, with the majority (n = 33/46, 71.7%) being females. Four patients within the *RAS*-wt cohort had multiple NIFTPs: one patient had five nodules, two subjects had three NIFTPs, and another patient had two. Thus, the median number of NIFTPs was 1 (range: 1–5) and the total number of nodules was 55. The mean nodule diameter was 17.7 ± 16.6 mm, with 25 nodules (45.5%) arising in the right lobe, 27 (49.1%) in the left lobe, and 3 (5.5%) in the thyroid isthmus.

Subjects in the *RAS*-mut group (n = 23 patients, 33.3%) were significantly younger (mean age 47.3 ± 15.5 years, *p* = 0.0034). A female predominance was noted also in this cohort (n = 19/23, 83%). The mean nodule diameter was 21.5 ± 13.9 mm, with 14 nodules (46.7%) arising in the right lobe, 14 (46.7%) in the left lobe, and 2 (6.7%) in the thyroid isthmus. *NRAS* was the most frequently mutated gene (n = 25, 83.3% of *RAS*-mut NIFTPs), followed by *HRAS* (n = 4, 13.3%) and *KRAS* (n = 1, 3.3%). Interestingly, the NGS analysis detected an *NRAS* Q61R mutation in a case where the corresponding immunohistochemical assay had yielded a negative result. Four patients within the *RAS*-mut group had multiple NIFTPs: three subjects had three nodules each, while another patient had two NIFTPs. Therefore, the median number of NIFTPs was 1 (range: 1–3) and the total number of nodules was 30, bringing the overall number of lesions analyzed in the study to 85.

### 2.2. Discriminative Protein Signatures Between RAS-mut and RAS-wt NIFTPs Identified by MALDI-MSI

The application of MALDI-MSI allowed the direct identification of NIFTPs (red- and yellow-colored regions) from the surrounding non-neoplastic thyroid parenchyma (green- and purple-colored areas) within each TMA core (Figure 1A). Indeed, bisecting K-means clustering yielded four distinct proteomic groups corresponding to two subsets of NIFTPs, surrounding thyroid parenchyma, and fibrotic regions, respectively, as highlighted by the score chart of their spectra (Figure 1B and Appendix A). Peptide spectra derived from the analysis of the annotated NIFTP regions revealed 349 aligned *m*/*z* values. By performing a receiver operator characteristic (ROC) analysis and applying a Student’s t-test with the Benjamini–Hochberg correction method for multiple comparisons on the above-mentioned peaks, we identified 92 *m*/*z* signals differentially expressed in RAS-mut compared to *RAS*-wt NIFTPs with AUC values of > 0.65 (Figure 2A; Appendix A). In a principal component analysis (PCA) performed to investigate the discriminative potential of these signals, the first three principal components explained 89.9% of the total data variability (PC1: 48.7%; PC2: 29%; PC3: 12.2%), thus showing that unsupervised statistical analyses yield proteomic signatures that discriminate between *RAS*-mut and *RAS*-wt NIFTPs (Figure 2B).

From the 92 statistically significant MALDI signals, we selected the 24 *m*/*z* peaks with the highest discriminatory potential (AUC ≥ 0.75. Marked with an asterisk in Appendix A). Interestingly, the intensity distributions of all peptides were significantly lower in *RAS*-mut as compared to *RAS*-wt NIFTPs, indicating that these peptides were all down-expressed in the former lesions (Figure 2C).

### 2.3. Identification of Differentially Expressed Proteins by MALDI-MS/MS

By combining MALDI-MSI with MALDI-MS/MS, we were able to putatively identify four differentially-expressed proteins, namely Serine/threonine-protein phosphatase 2A (PP2A) regulatory subunit B (*m*/*z* 957.5469 ± 20 ppm, P2R3A. Figure 3), Histone H2A type 1-A (*m*/*z* 944.5305 ± 20 ppm, H2A1A), Histone H4 (*m*/*z* 1325.7491 ± 20 ppm, H4), and ATP-dependent RNA helicase DDX42 (*m*/*z* 1542.7360 ± 20 ppm, DDX42) (Appendix A). These proteins are involved in the organization of chromatin structure (H2A1A and H4), regulation of mRNA splicing (DDX42), and modulation of cell growth (P2R3A) [10,11,12], suggesting a disruption in these key cellular processes as a potential pathogenetic mechanism specific to *RAS*-mut NIFTPs.

## 3. Discussion

Follicular-patterned thyroid neoplasms pose a significant differential diagnostic challenge due to their cytomorphological similarities and partially overlapping mutational profile [1]. Considering the profound differences in terms of clinical course and prognosis within this group of lesions, extensive research efforts have focused on the identification of biomarkers able to complement morphological and molecular genetic analyses, especially in borderline scenarios [13]. In many of such cases, the spectrum of differential diagnoses includes the NIFTP category, given the heterogeneous histological appearance and molecular profile of these tumors, which frequently harbor *RAS* alterations [3]. Thus, the introduction of easily testable markers is anticipated to improve the diagnostic performance in challenging situations for both surgical and cytological specimens [14]. The identified biomarkers may also provide insights into the pathogenesis of the lesions being studied, potentially suggesting intrinsic biological differences between tumors with similar histological appearances. This appears to be the case for NIFTPs, with preliminary reports by our group showing that *RAS*-wt NIFTPs closely resemble hyperplastic nodules in terms of their proteomic profile, while *RAS*-mut NIFTPs display an increased expression of RAS-interacting proteins, suggesting that constitutive RAS activation may play a pivotal role in the biology of the latter lesions [9]. In turn, this could endow *RAS*-mut NIFTPs with the hallmarks of *RAS*-driven tumors, such as increased survival and proliferative capacities. A preliminary clue that may support this hypothesis is the significantly earlier emergence of *RAS*-mut compared to *RAS*-wt NIFTPs, as observed in the present study.

Several other groups have taken advantage of proteomic analyses through MALDI-MSI to achieve an in-depth characterization of challenging thyroid lesions. For instance, it has been shown that NIFTPs and encapsulated follicular variants of PTC (EFVPTCs) share a proteomic signature that is distinct from that of classic PTC as well as of normal thyroid parenchyma [15]. The combination of MALDI-MSI with other “omic” techniques, particularly NGS, may further strengthen its potential for biomarker discovery by allowing the investigation of the relationship between genetic abnormalities and alterations in protein expression patterns [8,9]. In the present study, we employed this integrated proteogenomic methodology to confirm our previous findings that MALDI-MSI unsupervised segmentation enables the accurate discrimination of the NIFTP area from the surrounding non-neoplastic thyroid parenchyma. Additionally, we demonstrated significant differences between *RAS*-mut and *RAS*-wt NIFTPs in terms of their proteomic profiles, confirming the heterogeneity of the lesions included in the NIFTP category. Several proteins were found to be down-expressed in *RAS*-mut cases, four of which were identified in situ via MALDI-MS/MS. Interestingly, all these four proteins have been linked to the development and progression of neoplastic disease. One of them, namely the P2R3A regulatory subunit of protein phosphatase 2A (PP2A), is also closely related to the RAS-MAPK pathway. The serine/threonine phosphatase PP2A is a tumor suppressor that has been found to be altered in breast, skin, lung, and colorectal cancer [16]. The PP2A holoenzyme is a trimeric complex consisting of scaffold, catalytic, and regulatory subunits. The latter belong to four distinct families, namely B, B′, B″, and B‴, each comprising different isoforms. B subunits modulate the interactions between PP2A and its substrates and determine the intracellular compartmentalization of the holoenzyme complexes [17]. Interestingly, it has been shown that overexpression of B″ family subunits (which include the P2R3A protein identified in our study) halts progression through the cell cycle in vitro [16]. Suppression of P2R3A has also been implicated in malignant transformation through increased c-Myc signaling [18]. Thus, it could be argued that the reduced expression of P2R3A in *RAS*-mut NIFTPs may result in enhanced proliferative capacities as compared to their RAS-wt counterparts. Although the precise mechanisms linking RAS pathway aberrations and P2R3A loss remain to be elucidated, one possible explanation involves the role of PP2A in the assembly of the RAS signalosome. In quiescent cells, the PP2A catalytic and scaffold subunits are constitutively bound to the Raf-1 protein and the Kinase Suppressor of RAS 1 (KSR1)-mitogen-activated protein kinase (MEK) complex in the cytoplasm. Upon RAS activation in response to growth factors, the PP2A PR55 regulatory subunit (a member of the B family) binds to Raf-1 and to the KSR1-MEK complexes, promoting their dephosphorylation by PP2A. This event induces their translocation to the plasma membrane, where Raf-1 is activated, and KSR1 facilitates the activation of downstream mediators in the MAPK signaling cascade, such as MEK and ERK [19]. Based on this, we propose that the constitutive RAS activation in *RAS*-mut NIFTPs preferentially induces the expression of the PR55 regulatory subunit, leading to a compensatory downregulation of the P2R3A subunit.

As for the DDX42 RNA helicase, recent studies have elucidated its critical role in the assembly of the U2 small nuclear ribonucleoprotein particle (snRNP), which is a component of the human spliceosome [11]. DDX42 also has RNA chaperone activity, contributing to the correct folding of RNA molecules [20]. Thus, it is likely that DDX42 downregulation would result in altered pre-messenger RNA (pre-mRNA) processing. Since abnormal splicing regulation has been closely linked to tumor cell proliferation and invasiveness [21], one might argue that DDX42 downregulation could endow *RAS*-mut NIFTPs with enhanced proliferative capacities and a tendency for more rapid growth.

Lastly, histones play crucial roles in chromatin organization and stability, as well as in the regulation of gene expression and centromere function [10]. In the nucleosome structure, the histone octamer around which DNA is wrapped is composed of two copies of each of the four core histones, namely H2A, H2B, H3, and H4. Experimental evidence supports the role of aberrant histone modifications, such as methylation and acetylation, in tumor development and progression [22]. Therefore, the observed down-expression of H2A1A and H4 histones in *RAS*-mut NIFTPs could be ascribed to an increased burden of post-translational modifications as compared to *RAS*-wt lesions. This, in turn, could reflect widespread alterations in gene expression in the former group of lesions. An alternative explanation for H2A1A downregulation could involve the increased expression of other histone variants, similar to what has been described in other neoplasms. For instance, overexpression of the histone H2A variant H2A.X in breast cancer promotes metastasis [23], and increased H2A.X mRNA levels have been identified as a negative prognostic factor [24].

It is interesting to note that the number of differentially expressed signals reported in the present work was greater than in our previous study on NIFTPs [9], thanks to an improved mass spectrometry workflow enabled by the more sensitive and accurate timsTOF fleX mass spectrometer. Moreover, the proteins identified herein were not among the differentially expressed signals previously reported by our group. This further highlights the marked heterogeneity within the NIFTP category, stressing the need for a panel of markers rather than a standalone assay to aid in the differential diagnosis between challenging follicular lesions. Nevertheless, in both studies, the identified protein signature was notable for the presence of RAS-interacting proteins (e.g., cyclophilin A in our initial work and P2R3A in the current study), supporting the role of the MAPK pathway as a key determinant of the biology of *RAS*-mut NIFTPs [9].

Another noteworthy aspect is that none of the genes encoding the proteins identified in both our studies have been consistently associated with thyroid neoplasms. Indeed, the most widely employed multigene classifiers for the triage of indeterminate thyroid nodules (e.g., ThyroSeq^®^ v3, Afirma^®^ Gene Expression Classifier [Afirma^®^-GEC], ThyGeNEX^®^/ThyraMIR^®^) do not test for such genes [25,26,27]. This highlights the ability of proteomic analyses to complement genomic data, allowing the identification of novel markers that could support the diagnostic process.

In conclusion, we have identified four proteins that are significantly downregulated in *RAS*-mut NIFTPs, warranting further validation as potential biomarkers to assist in the diagnostic process of challenging thyroid lesions. Further studies are required to assess whether the downregulation of the four proteins is a distinctive feature of *RAS*-mut NIFTPs or is a hallmark of all follicular tumors with *RAS* mutations. If that were the case, documentation of a reduced expression of these proteins may serve as a surrogate indicator of the presence of a *RAS* mutation in a follicular-patterned neoplasm. Thus, it would provide a cost-effective tool to document the *RAS* mutational status of a follicular tumor in cases where molecular genetic analyses are not available or have yielded indeterminate results. Importantly, the proposed panel could possibly retain its diagnostic value even in the presence of less common *RAS* alterations, which are not covered by the currently available *NRAS* Q61R immunohistochemical assay. If, on the other hand, the identified proteins were found to be conserved in *RAS*-mut follicular lesions other than NIFTPs, documenting their retained expression may help to specifically rule out a NIFTP diagnosis in morphologically borderline tumors with *RAS* mutations. Early experience from our group suggests that DDX42 and histone H4 expression is indeed conserved in FAs and FVPTCs (unpublished data), providing a preliminary confirmation of the differential diagnostic value of the identified markers.

Finally, the four-protein signature described herein could also reflect unrecognized pathobiological mechanisms specific to *RAS*-mut NIFTPs, which critically depend on the presence of *RAS* mutations. This suggests that, despite their morphological similarities, *RAS*-mut and *RAS*-wt NIFTPs may be intrinsically distinct entities, with the former showing *RAS*-dependent disruptions in key cellular processes. Therefore, additional prospective studies with extended follow-up may be warranted to investigate potential differences in natural history and clinical behavior (e.g., risk of contralateral recurrence after lobectomy) between the two NIFTP subcategories.

## 4. Materials and Methods

### 4.1. Histopathology and TMA Preparation

Formalin-fixed, paraffin-embedded (FFPE) samples and the corresponding diagnostic slides of 85 NIFTPs were collected from the archives of the Department of Pathology, Fondazione IRCCS San Gerardo dei Tintori (University of Milano-Bicocca-UNIMIB), Monza, Italy. The study was approved by the local ethical committee (FINAL-TIR PU 3581/21).

Hematoxylin and eosin (H&E)-stained slides were reviewed by two expert thyroid pathologists (F.P. and V.L.) to confirm the NIFTP diagnosis based on the 2022 WHO classification criteria [28]. Specific regions of interest (ROIs) were selected for the tissue-microarray (TMA) construction. The ISE Galileo TMA R4.30 software (Integrated Systems Engineering, Milan, Italy) was employed to build the TMA layout. The preparation of the TMA blocks was made on a ISE Galileo TMA CK4500 semi-automated arrayer (Integrated Systems Engineering, Milan, Italy). One mm diameter cores were extracted from FFPE blocks, excluding areas containing artifacts or necrosis, as previously described [29]. Samples were divided into five sets and the corresponding TMA blocks containing 24, 50, 48, 24, and 24 cores, respectively, were prepared.

### 4.2. NGS

Nucleic acids for NGS analysis were extracted from 10 unstained slides with 4 µm-thick sections from each FFPE tissue block after manual microdissection with a sterile scalpel to increase tumor cell yield. DNA extraction was performed with Maxwell RSC DNA FFPE Kit (Promega, Madison, WI, USA) as per the manufacturer’s instructions. DNA quantification occurred on a Quantus Fluorometer (Promega) using the QuantiFluor ONE dsDNA System (Promega). Mutational analysis was carried out with the NGS panel Oncomine Comprehensive Assay (OCA) v3 System (ThermoFisher Scientific, Waltham, MA, USA), assessing insertions/deletions, copy-number variations (CNVs), and single-nucleotide variants (SNVs) of 161 cancer-associated genes. The full list of included genes can be retrieved at https://www.Thermofisher.com/order/catalog/product/A35805 (accessed on 2 August 2024). For library preparation, 10 ng of genomic DNA was automatically loaded onto the Ion 540 chip (ThermoFisher Scientific) using the Ion Chef System (ThermoFisher Scientific). Sequencing occurred via the Ion S5 System (ThermoFisher Scientific), and data analysis was performed using Ion Reporter Software v.5.16 (ThermoFisher Scientific). The median absolute pairwise difference (MAPD) parameter was used to identify low-quality samples at risk of generating false results. Specifically, cases with MAPD ≥ 0.5 were excluded from the analysis. Only mutations with appropriate quality metrics and an allele frequency ≥ 5% were reported. Data from three publicly available cancer genomics datasets (i.e., Catalogue of Somatic Mutations in Cancer, COSMIC, https://cancer.sanger.ac.uk/cosmic; cBioPortal, https://www.cbioportal.org/; and ClinVar, https://www.ncbi.nlm.nih.gov/clinvar/. Accessed on 2 August 2024) were used to identify pathogenic mutations. The Integrative Genomics Viewer (IGV) software v.2.16.0 (Broad Institute, Cambridge, MA, USA) was used to visually inspect clinically relevant and borderline mutations. For the identification of somatic variants, the filter settings included a total read depth of at least 1000×.

### 4.3. Sample Preparation for MALDI-MSI

The preparation of samples for the MALDI-MSI analysis was performed as previously described [30]. Briefly, 4 µm thick sections obtained from FFPE samples were mounted onto conductive indium tin oxide (ITO) slides. The latter underwent toluene dewaxing and tissue rehydration with decreasing concentrations of ethanol and water, followed by antigen retrieval by means of heating in a water bath containing 10 mM citric acid buffer at 97 °C, pH 5.95 for 45 min. On-tissue protein digestion was performed by trypsin deposition using an iMatrixSpray automated spraying system (Tardo Gmbh, Subingen, Switzerland) with the following configuration: heat-bed temperature: 37 °C; number of spray cycles: 15; enzyme density: 1.2 µL/cm^2^; movement speed: 160 mm/s; distance between spray lines: 2 mm; needle height: 45 mm. After overnight incubation at 40 °C in a humidity chamber, a matrix deposition for the MALDI–MSI analysis was performed by spraying α-cyano-4-hydroxycinnamic acid (10 mg/µL in 70% ACN, 30% H_2_O and 1% TFA) with an HTX TM-Sprayer (HTX Technologies, LLC, Chapel Hill, NC, USA) (temperature 75 °C; number of passes 4; flow rate 0.12 mL/min; velocity 1200 mm/min; track spacing 2 mm; pressure 10 psi).

### 4.4. MALDI-MSI Analysis

All TMA imaging analyses were performed using a timsTOF fleX mass spectrometer (Bruker Daltonics, Bremen, Germany) equipped with a Smartbeam™ 3D laser. The acquisition was performed in the *m*/*z* range of 700–3000. External calibration was carried out using a mixture of standard peptides (PepMix I, Bruker Daltonics) with a mass range *m*/*z* of 750–3150, which was directly added onto the ITO slides. The measurement regions were conducted using a raster width of 20 × 20 (x, y) and a lateral laser scan configuration of 16 μm. The instrument parameters for the acquisition method were configured using the timsControl software v.2.0.51.0_9669_1571 (Bruker Daltonics), and data visualization was performed using the FlexImaging software v.5.1 (Bruker Daltonics). After the MALDI–MSI analysis, the matrix was removed with increasing concentrations of ethanol (70% and 100%), and the slides were stained with H&E. The slides were converted to a digital format using a MIDI II digital scanner (3DHISTECH, Budapest, Hungary), enabling the overlap of the images and the integration of the morphological and proteomic data. The regions of interest (ROIs) of each TMA core were annotated by the pathologists.

### 4.5. In Situ Identification with MALDI-MS/MS Analysis

For in situ identification, MALDI-tandem mass spectrometry (MALDI-MS/MS) spectra were acquired using a precursor isolation window of ±2 Da. The MS/MS spectra were pre-processed using the FlexAnalysis 3.4 and BioTools 3.2 software (Bruker Daltonics). Baseline correction (TopHat algorithm) and spectral smoothing (Savitzky–Golay algorithm) were applied. All acquired MS/MS spectra were subjected to database searching against the Swiss-Prot human protein database utilizing the Mascot 2.8.3 search engine (Matrix Science, London, UK). Trypsin was set as a proteolytic enzyme, allowing for up to one missed cleavage. The search parameters included the oxidation of methione and FFPE methylol and methylene modifications. Mass tolerances were set lower than 0.2 Da for the precursor and at 0.5 Da for the fragments. The resulting protein identifications were statistically significant based on *p*-value thresholds and controlled false discovery rates (FDR).

### 4.6. Statistical Analysis

Mean (±standard deviation, SD) was calculated to describe continuous variables, while qualitative variables were reported as count and frequency. Between-group comparisons were performed by means of Student’s t-test or Wilcoxon rank-sum test (for quantitative variables), and Fisher’s exact test or chi-square test (for qualitative variables).

Concerning MALDI-MSI analysis, data files containing the individual spectra of each ROI were imported into SCiLS Lab 2024 Pro software (Bruker, Bremen, Germany). After baseline subtraction (TopHat algorithm), normalization (Total Ion Current algorithm), and spatial denoising, the average (avg) spectra, representative of the whole measurement regions, were generated. Subsequently, the previously annotated ROIs were defined on SciLS Lab software. Peak picking and alignment were performed for feature extraction, resulting in the detection of 349 *m*/*z* features. In addition, a Bisecting K-means algorithm was performed whereby the individual spectra showing a similar shape were grouped and uniformly colored. Principal component analysis (PCA) was performed on the individual spectra to achieve a dimensionality reduction and to show the data in a 3D setting. In the subsequent receiver operating characteristic (ROC) curve analysis, a peak was considered statistically significant if the corresponding area under the curve (AUC) and *p*-value (calculated with Student’s t-test with the Benjamini–Hochberg correction method for multiple comparisons) were ≥0.75 and ≤0.001, respectively. For mass spectra visualization, the open-source software mMass v.5.5 (http://www.mmass.org. Accessed on 8 July 2024) was used [31,32]. Statistical analyses were performed using the open-source SciLS Lab and MetaboAnalyst 6.0 software applications [33].

## Figures and Tables

**Figure 1 ijms-25-13115-f001:**
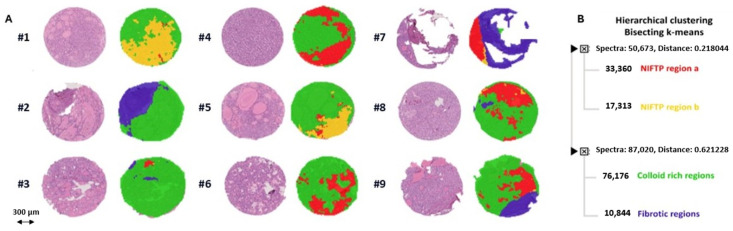
Automatic segmentation of TMA cores by MALDI-MSI. (**A**) Comparison of MALDI-MSI segmentation images with the corresponding H&E image of a representative cohort of TMA cores. (**B**) Distinction of NIFTPs (red- and yellow-colored regions) from the surrounding thyroid parenchyma (green-colored areas) and fibrotic areas (purple) through bisecting K-means clustering of the TMA cores based on the proteomic MSI data. H&E, hematoxylin and eosin; MALDI-MSI, matrix-assisted laser desorption/ionization mass spectrometry imaging; NIFTP, noninvasive follicular thyroid neoplasm with papillary-like nuclear features; TMA, tissue microarray.

**Figure 2 ijms-25-13115-f002:**
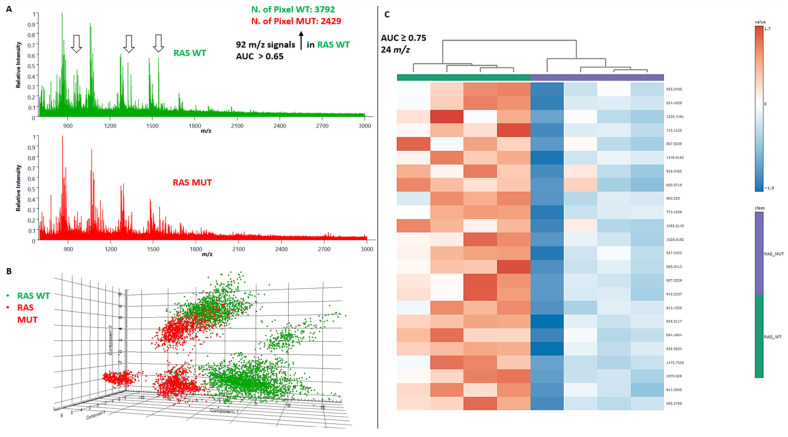
Discriminative protein signatures between *RAS*-mut and *RAS*-wt NIFTPs identified by MALDI-MSI. (**A**) Peptide spectra derived from the analysis of the annotated NIFTP regions. (**B**) PCA score plot of the individual spectra. (**C**) Comparative heatmap and hierarchical clustering of the expression levels of the 24 *m*/*z* signals with the highest discriminatory potential (AUC ≥ 0.75) for *RAS*-mut vs. *RAS*-wt NIFTPs. AUC, area under the curve; MALDI-MSI, MALDI-MSI, matrix-assisted laser desorption/ionization mass spectrometry imaging; mut, mutant; NIFTP, noninvasive follicular thyroid neoplasm with papillary-like nuclear features; PCA, principal component analysis; wt, wild-type.

**Figure 3 ijms-25-13115-f003:**
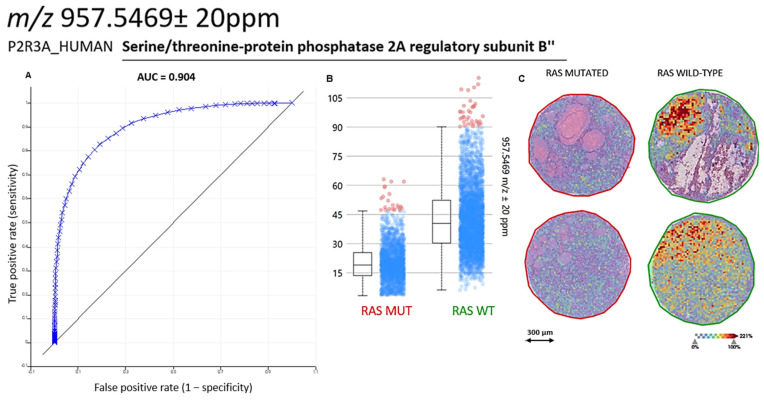
Comparison of P2R3A expression in *RAS*-mut vs. *RAS*-wt NIFTPs. (**A**,**B**) AUC and intensity box plots for the comparison of P2R3A expression in *RAS*-mut vs. *RAS*-wt NIFTPs. (**C**) MALDI-MSI images showing the spatial localization of the P2R3A signal (*m*/*z* 957.5469 ± 20 ppm) in two different *RAS*-mut NIFTPs (**left**) and in two different *RAS*-wt NIFTPs (**right**). A scale bar on the bottom left is shown, as well as a color-coded scale for signal intensity. AUC, area under the curve; MALDI-MSI, matrix-assisted laser desorption/ionization mass spectrometry imaging; mut, mutant; NIFTP, noninvasive follicular thyroid neoplasm with papillary-like nuclear features; P2R3A, Serine/threonine-protein phosphatase 2A regulatory subunit B″; wt, wild-type.

**Table 1 ijms-25-13115-t001:** Summary of clinicopathologic features of the study cohort.

Clinicopathologic Parameter		*RAS*-WT (n = 46 pts)	*RAS*-MUT (n = 23 pts)	Overall (n = 69 pts)	*p*-Value
**SEX**	**M**	13 (28.3)	4 (17)	17 (24.6)	0.3233
	**F**	33 (71.7)	19 (83)	52 (75.4)	
**AGE (years)**		57.3 ± 11.4	47.3 ± 15.5	54.0 ± 13.7	0.0034
**NODULE DIAMETER (mm)**		17.7 ± 16.6	21.5 ± 13.9	19.0 ± 15.8	0.3486
**N° OF NIFTPs**	**1**	42 (91.3)	19 (82.7)	61 (88.4)	0.6384
	**2**	1 (2.2)	1 (4.3)	2 (2.9)	
	**3**	2 (4.3)	3 (13.0)	5 (7.2)	
	**5**	1 (2.2)	0 (0)	1 (1.4)	
**LOBE ^**	**R**	25 (45.5)	14 (46.7)	39 (45.9)	0.9616
	**L**	27 (49.1)	14 (46.7)	41 (48.2)	
	**I**	3 (5.5)	2 (6.7)	5 (5.9)	

Data reported as N (%) for categorical variables; mean ± SD for continuous variables. ^ Number of NIFTPs for each lobe. F, female; I, isthmus; L, left; M, male; MUT, mutated; N, number; pts, patients; R, right; SD, standard deviation; WT, wild-type.

## Data Availability

Data that support the findings of this study are available upon reasonable request from the corresponding author V.L.

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
