# Peer review of "Spatially Resolved Molecular Characterization of Noninvasive Follicular Thyroid Neoplasms with Papillary-like Nuclear Features (NIFTPs) Identifies a Distinct Proteomic Signature Associated with RAS-Mutant Lesions"

_ijms, 2024, doi:10.3390/ijms252313115_

Round 1
Reviewer 1 Report
Comments and Suggestions for Authors
In this manuscript the authors analyze a particular subtype of thyroid tumors called noninvasive follicular thyroid neoplasms with papillary like nuclear features (NIFTP), in particular those wild type and mutated for the RAS gene, by means of a proteomic approach that uses MALDI-MSI.
The aim of the work is basically to identify by proteome a signature of biomarkers that can be used for the diagnosis of this thyroid tumor subtype. The work is very well done, only some points need to be better clarified.
Regarding the first paragraph of the Results, I would like to ask the authors to describe and explain Table 1 better, as it is not very clear how many patients and how many NIFTIP are analyzed, especially the final part of the table. When the NIFTP present in the two lobes of the thyroid are described the manuscript and Table 1 become misleading. Please clarify this aspect better. Likewise, it is not very clear and it is necessary to better explain how many and how the RAS gene mutations are distributed in these NIFTPs.
As for paragraph 2.3, I would like to ask the authors to cite the DDX42 protein in the text of the manuscript, saying that it is shown as an example of a protein identified in Figure 3.
Furthermore, as for Figure 3 and the subsequent supplementary Figures, I would like to ask the authors to include in the caption regarding panel C, the fact that two different mutated cases and two wild type cases for RAS are shown respectively, in which, if necessary, an identification code can also be inserted.
A general question regarding the manuscript is the following: why do the authors only identify under-expressed proteins in the mutated RAS NIFTPs compared to the wild type?
Finally, I would like to ask the authors to describe in more detail, both in the Introduction and in the Discussion, the differences and improvements obtained with respect to their first reported work on the present topic published in 2023 on IJMS (see reference n 9).
Author Response
Please see the attachment.
Changes to the manuscript text are highlighted in yellow in the revised version of the manuscript.
To enhance clarity, an additional change has been made to Figure 1: panel C has been removed and is now included as Supplementary Figure S1 in the revised manuscript.

Reviewer 2 Report
Comments and Suggestions for Authors
The manuscript entitled “Spatially resolved proteogenomic characterization of Noninvasive Follicular Thyroid Neoplasms with Papillary-Like Nuclear Features (NIFTPs) identifies a distinct proteomic signature associated with RAS-mutant lesions” by Denti et al. described finding four peptides to distinguish RAS-mut and RAS-wt NIFTPs by using MALDI-MSI. It is interesting to discovery the proteomic features that are relevant to RAS-mut tumors, however, there are some concerns about the study.
1. The title of the paper mentioning “proteogeonomics,” however, only proteomics data was utilized in the study. The only thing seems related to “genomics” that was when authors described samples with RAS-mut or RAS-wt. When talking about proteogenomics, the authors should conduct analyses showing, for example, what they can find using genomics/transcriptomics, which could also be found in proteomics, or what they can identify from proteomics that genomics/transcriptomics cannot etc. Recently, there are many papers describing proteogenomic characterization for various cancer types, perhaps, the authors will find these papers useful in some way. On the other hand, if proteomics is the central theme of the study, perhaps, the authors should not state “proteogenomics characterization” in the title.
2. How did the authors perform ROC analysis? Did authors use cross-validation? Since the sample size is quite small in this study, the ROC curves seem overfitting and the high AUCs might be occurred by chance. The authors should validate their findings by using an independent cohort.
3. Did the authors find 4 peptides or 4 proteins as significant features? It seems that the authors used peptides and protein interchangeably. However, proteins and peptides are different. Did the authors actually identify the proteins, namely, H2A1A, P2R3A, H4, DDX42, or the peptides from these 4 proteins? The authors need to clarify in the paper and make sure they use the correct terminology.
4. The authors mentioned about “92 m/z signals differentially expressed in RAS-mut as compared to RAS-wt NIFTPs…” Did the authors perform any statistical analysis and multiple testing correction?
Author Response

(The authors gave the same response as above.)

Round 2
Reviewer 2 Report
Comments and Suggestions for Authors
I would like to thank the authors for addressing my comments and revising the manuscript. I have no further comments and suggestions.